# The Relationships between the Structure and Properties of PA56 and PA66 and Their Fibers

**DOI:** 10.3390/polym15132877

**Published:** 2023-06-29

**Authors:** Keming Luo, Jiaxin Liu, Kieth Abbay, Yangjie Mei, Xiaowei Guo, Yunhe Song, Qingbao Guan, Zhengwei You

**Affiliations:** 1State Key Laboratory for Modification of Chemical Fibers and Polymer Materials, College of Materials Science and Engineering, Institute of Functional Materials, Research Base of Textile Materials for Flexible Electronics and Biomedical Applications (China Textile Engineering Society), Shanghai Engineering Research Center of Nano-Biomaterials and Regenerative Medicine, Donghua University, Shanghai 201620, China; 2220588@mail.dhu.edu.cn (K.L.); 2220564@mail.dhu.edu.cn (J.L.); 2210534@mail.dhu.edu.cn (K.A.); 2210423@mail.dhu.edu.cn (Y.M.); 2Heilongjiang EPPEN New Materials Co., Ltd., Daqing 166299, China; guoxiaowei@eppen.com.cn (X.G.); songyunhe@eppen.com.cn (Y.S.)

**Keywords:** polyamide 56, 1,5-pentanediamine, bio-based, biodegradability, flame retardancy

## Abstract

Bio-based polymers can reduce dependence on nonrenewable petrochemical resources and will be beneficial for future sustainable developments due to their low carbon footprint. In this work, the feasibility of bio-based polyamide 56 (PA56) substituting petroleum-based PA66 is systematically investigated. The crystallization, melting, and decomposition temperature of PA56 were all lower than that of PA66. PA56 formed a γ crystal type with larger grain size and took a longer amount of time to complete the crystallization process since its crystallization rate was lower than that of PA66. Compared with PA66, PA56 exhibited a higher tensile strength of 71.3 ± 1.9 MPa and specific strength of 64.8 ± 2.0 MPa but lower notched impact strength. More importantly, the limited oxygen index and vertical combustion measurement results indicated that the flame retardancy of PA56 was better than PA66, and the LOI values and the UL94 result of PA56 were 27.6% ± 0.9% and V-2. It is worth noting that the PA56 fiber had superior biodegradability compared to the PA66 fiber. PA56 showed significant biodegradation from the eighth week, whereas PA66 remained clean until the sixteenth week (without obvious biodegradation taking place). Eventually, PA56 did not show significant differences compared to PA66 in terms of thermal and mechanical properties. However, PA56 had great advantages in flame retardancy and biodegradability, indicating that the bio-based PA56 could potentially replace petroleum-based PA66 in many fields.

## 1. Introduction

Recently, renewable polymers from biomass have attracted significant attention all over the world [1]. These bio-based polymers can reduce dependence on nonrenewable petrochemical resources [2], decrease greenhouse gas emissions, and eventually provide environmentally and socially sustainable developments, leading to a low-carbon-footprint society [3]. Polyamide 66 (PA66) is a versatile thermoplastic with a wide range of applications, including engineering plastics [4], fibers [5], and films [6,7], because of its good physical and mechanical properties [8]. To date, it is still difficult to produce bio-based PA66 without bio-sourced monomers, 1,6-hexamethylene diamine [9,10]. The commercialized 1,5-Pentanediamine, obtained via lysine fermentation [11], could be combined with adipic acid to fabricate a partly bio-based polyamide-PA56. Due to the fact that PA56 has a similar structure and properties to that of PA66, PA56 is considered a promising bio-based alternative to PA66.

The preparation, characterization, and modification of PA56 have all been reported in last decade [12,13]. For example, Eltahir et al. successfully utilized a combined approach and described the non-isothermal crystallization process [14]. Eitahir et al. explored the effect of a hot drawing process at different draw ratios and temperatures on the properties of PA56 fiber [15]. Zhang et al. achieved ultra-toughening by improving the interface compatibility of PA56 through twin-screw extrusion [16]. Recently, PA56 has been processed into fibers, films, and membranes [17]. For instance, Xue et al. proposed a green bioprocess for the fabrication of nanofiber membranes from the bio-based PA56 via electrospinning [18]. Liu et al. described a strategy to construct bio-based PA56 nanofiber membranes with bimodal structures for effective air filtration via one-step electrospinning/netting [19]. Puiggalí and Morales-Gámez analyzed the crystal structure of PA56 and investigated the irreversible Brill transition and two different bounding directions of PA56, respectively [20].

Numerous efforts have been made to develop degradable or recyclable polymers by introducing various natural fillers or monomers [21,22]. Synthetic polymers, including nylon, are a growing environmental concern because they are generally nondegradable, and recent research has focused on the biological treatment of plastic wastes and the development of biodegradable plastics [23]. Although the similarities and differences between PA56 and PA66 have been investigated, their biodegradability are still not fully understood. It is interesting and useful to make a systematic comparison of the integrated properties between PA56 and PA66, especially in terms of their biodegradability, to provide solutions to facilitate green, low-carbon sustainable developments [24]. For this article, we performed statistical tests to compare the properties of PA56 and PA66. The results reported herein indicate that the bio-based PA56 could potentially replace petroleum-based PA66 in many fields (Figure 1).

## 2. Experimental Section

### 2.1. Materials

PA66 and bio-based PA56 pellets were kindly supplied by EPPEN Biotech Co., Ltd. (Yinchuan, China). 1,1,1,3,3,3-hexafluoro-2-propanol (HFIP) was purchased from TCI (Shanghai, China). All the chemicals were used without further purification. 

### 2.2. Characterizations

Fourier-transform infrared spectroscopy

Infrared spectra of PA56 and PA66 were determined using a Nicolet 6700 FT-IR spectrophotometer (Thermo Scientific, Waltham, MA, USA) with an attenuated total reflection (ATR) attachment. Wavenumber range from 4000 cm^−1^ to 400 cm^−1^.

Thermogravimetric analysis

Thermogravimetric analysis (TGA) was carried out using Discovery TGA (TA Instruments, New Castle, DE, USA) at a heating rate of 10 °C/min under N_2_ (20 mL/min). Test temperatures from 40 °C to 600 °C.

Differential scanning calorimetry

Differential scanning calorimetry (DSC) was performed on Netzsch DSC 204 Phoenix (Bayern, Germany) under nitrogen atmosphere with a gas flow rate of 60 mL/min. The data were collected in a range of 40–310 °C at a heating rate of 10 °C/min. Crystallization behavior at different cooling rates, including 20, 10, and 5 °C/min, were also evaluated via DSC.

Wide-angle X-ray diffraction

Wide-angle X-ray diffraction (WAXD) analysis was conducted on a Rigaku D/max-2550 PC (Tokyo, Japan) at room temperature. The Cu/Kα radiation (λ = 1.54 Å) was operated at 40 kV and 200 mA. The 2θ scan data were collected at 0.02° intervals over the range of 3°–45°.
Bragg equation nλ = 2d∙sinθ(1)
where λ (nm) is the wavelength of the X-rays, n is the number of diffraction levels, d (nm) is the crystal plane spacing, and θ is the angle between the incident X-rays and the corresponding crystal plane [25].
Scherre equation D_bkl_ = Kλ/(β∙cosθ)(2)
where K is a constant, which depends somewhat on particle shape but is usually about 0.9; λ (nm) is the wavelength of the X-rays; β (nm) is the angular width of the beam at which the intensity falls to half its maximum value; and D_bkl_ measures the average dimension of the particles in a direction perpendicular to the Bragg planes giving rise to the reflection [26].

Melt index test

Melt index test was carried out to evaluate the melt flow properties. ASTM standard D 1238 was followed for the measurements, with barrel temperature of 275 °C and load of 0.325 kg (as recommended).

Rheological test

Rheological test was performed using a double-tube capillary rheometer (Rosand RH2000, Netzsch, Bayern, Germany ) with a capillary (R = 0.5 mm, L/D = 32) under 275 °C.

Gel Permeation Chromatography test

The relative molecular weights and distribution of PA56 were determined via Malvern gel chromatography (Worcestershire, UK) with high-performance liquid chromatography grade hexafluoroisopropanol as solvent and PMMA polymers with molecular weights of 1000~300,000 as standard; PA56 and PA66 were prepared as 0.1 mg/mL solution, fully dissolved, and then filtered for testing.

Tensile and flexural properties test

Tensile and flexural properties were measured according to the national standard GB/T 1040-92 and GB/T 9341-2000 at testing speeds of 15 mm/min and 1 mm/min, respectively. The V-notches impact strength with diameter of 2.5 ± 0.05 cm was measured under 15 °C according to the standard GB/T 1843-2008/ISO180:2000.

Combustion performances test

Combustion performances of the polymers were measured by means of testing limiting oxygen index (LOI) on the basis of national standard (GB/T 2406.2–2009/ISO 4589-2:1996) and vertical flammability (UL94) on the basis of national standard (GB/T 2408-2008/EIC 60695-11-10:1999).

Biodegradability test

The biodegradability of PA56 and PA66 fibers were tested in a culture fluid enriched with phanerochaete chrysosporium and by extracting the samples from the culture fluid at regular intervals to measure the molecular weight and distribution.

Scanning Electron Microscope

Surface morphology images of PA56 and PA66 fibers after biodegradation were tested by using a Scanning Electron Microscope (SEM). Fibers were sprayed gold 120 s before testing.

## 3. Results and Discussion

### 3.1. Structural Characterization

FT-IR spectra revealed the similar structures of PA56 and PA66, with characteristic absorption at 1462 cm^−1^/1461 cm^−1^ (CH_2_ symmetric scissors deformation), 1543 cm^−1^/1539 cm^−1^ (N-H bend and C-N stretch), 1637 cm^−1^/1637 cm^−1^ (amide C=O stretch), 2859 cm^−1^/2859 cm^−1^ (CH_2_ symmetric stretch), 2930 cm^−1^/2931 cm^−1^ (CH_2_ asymmetric stretch), 3082 cm^−1^/3080 cm^−1^ (NH bend overtone), and 3300 cm^−1^/3307 cm^−1^ (NH stretch), respectively (Figure 1) [27].

### 3.2. Crystallization Behavior and Thermal Properties

Our wide-angle X-ray diffraction (WAXD) analysis provided detailed information and revealed the fine crystal structure of the polymers. PA66 showed a broad characteristic peak that was formed by two overlapped peaks (Figure 2). The fully extended polymer chain of PA66 mainly form α-form crystals, and the antiparallel molecular chains form hydrogen bonds in the (002) plane. However, PA56 showed a strong sharp peak at 2θ = 20.48, which was assigned to the (020) plane. The conventional α/β forms usually found in commercial even-even PAs (e.g., PA66), which cannot be expected for the odd-even structure of PA56, considering the minimization of packing energy. A pseudohexagonal structure, the WAXD pattern of which was characterized by a single strong equatorial reflection at d-spacing was approximately 4.15 Å, had been postulated for PAs derived from an odd diamine and/or an odd dicarboxylic acid (e.g., PA56) [20]. In contrast to PA 66, the crystallinity of PA56 was relatively lower; however, the crystal size was larger. The d-spacing values, crystalline size, and crystallinity were calculated by employing the Bragg equation and Scherre equation, respectively. The corresponding data are summarized in Table 1.

Non-isothermal crystallization process was performed on PA56 and PA66 using DSC analysis at different cooling rates. Both the crystallization onset and endset temperature of PA56 were lower than that of PA66, as shown in Figure 3 and summarized in Table 2. The odd-odd structure of PA66 led to a higher H-bond content among polymer chains, which would require more energy for the polymer chains to move and fold into an ordered crystal structure. As the cooling rate increased, the crystallization peak of PA56 moved to a lower temperature [28]. Meanwhile, the peak shape became broader, indicating that the polymer chains need more time to finish the crystallization process. It can be understood that the polymer chains have enough time to adjust the configuration at a lower cooling rate and can be perfectly allied and densely packed at a high temperature, which also made the crystallization process shorter and the peak sharper. Additionally, at a lower temperature, the mobility of molecules was limited, and the larger supersaturation accelerated the nucleation process. Meanwhile, the crystal growth was faster, making the crystallization more imperfect and the peak broader.

Our DSC analysis revealed that the melting temperature of PA56 (252.9–253.9 °C) was slightly lower than that of PA66 (258.2 °C), as shown in Figure 4 and summarized in Table 3. In contrast to the single melting peak of PA66, there was an exothermic process prior to the melting process for PA56, likely ascribed to the Brill transformation from a pseudohexagonal crystal structure to a monoclinic lattice. All three PA56 samples have a similar melting process, with initial melting occurring at ~246 °C and an endset temperature at ~258 °C, and the whole melting process was conducted in a range of about 11 °C.

TGA analysis (Figure 5) revealed that the initial decomposition temperature (T_d_^5%^) of PA56 was about 414 °C, which was slightly lower than that of PA66 (430 °C). The temperature of maximum degradation (T_dmax_) of both PA56 and PA66 was about 478 °C.

### 3.3. Melt Rheology

Melt flow index (MFI) plays an important role in the plastics industry and is a useful tool for quality control and assurance [29]. In particular, MFI fulfills the requirements for checking the quality of polymers and assessing its processability at low shear rate. The MFI of PA56 was higher than that of PA66 due to the relatively lower molecular weight of PA56 (Table 4), the molecular chains of which flowed easily and suffered less friction. This implies that PA56 might have even better processability than PA66, enabling it to be fabricated into continuous and uniform textile fibers, plastics, and films.

Polymer melts exhibit elastic and viscous properties, so the rheological measurements of materials are important for optimizing processing conditions such as extrusion, injection molding, and melt spinning, which are highly dependent on temperature, pressure, molecular weight, and shear rate [30]. However, more customized measurements are needed since the polymer melt suffered high shear stress during the high-speed melt spinning process, meaning that the melt index test is no longer sufficient for reflecting accurate results. Both PA56 and PA66 exhibited the pseudoplastic fluid features of shear thinning, and their shear viscosity decreased at different amplitudes as the shear rate increased (Figure 6). The shear stress and shear rate of PA56 were much lower than PA66 in the same conditions (temperature and shear rate, etc.). The non-Newtonian index (n) of pseudoplastic fluid to some extent indicates the dependence of shear viscosity and shear stress on shear rate. The n value can be obtained by the power-law equation.
η_α_ = Kγ^n−1^
where η_α_ (Pa·s), n, γ (s^−1^), and K represent shear viscosity, non-Newtonian index, shear rate, and constant, respectively [31].

When the non-Newtonian index approaches 1, the polymer is closer to Newtonian polymers. Both the shear viscosity and shear stress of the polymer are less sensitive to shear rate. The n values of PA56 and PA66 were 0.77 and 0.70, respectively. This indicates that the shear viscosity and shear stress of PA56 are less sensitive to shear rate compared to PA66, suggesting that PA56 is less likely to suffer from the Barus effect during the extruding process.

### 3.4. Mechanical Properties

Mechanical properties are important for practical application. The tensile strength, yield strength, and specific strength of PA56 were all higher than that of PA66, as shown in Figure 7A and Table 5. The structural similarities between PA56 and PA66 enable PA56 to inherit PA strength, such as flexural strength (Figure 7B) and hardness, while the yield strain and impact strength of PA56 were lower than that of PA66. The lower methylene content and fewer hydrogen-bonding interactions in PA56 reduced the flexibility of the polymer chain, which decreased the impact strength of the polymer. In addition, PA56 was less likely to undergo a large elongation process during the tensile test when compared to PA66, and the stress tended to reach the highest value at the yield point. This indicates that aliphatic PA56 derived from odd diamines and even dicarboxylic acids could not establish all the possible intermolecular hydrogen-bonding interactions when the molecular chains had an all-trans conformation, which would generate differences in mechanical properties [28].

The higher tensile strength and lower yield strain of PA56 contributed to its mechanical strength and dimensional stability, which could help broaden the range of nylon applications in engineering fields, such as in the automotive industry. However, the tensile properties of PA56 are far from sufficient to meet the requirements for the engineering of plastics. Currently, pultruded profiles are widely applied as structural elements in different areas due to their high strength, low weight, and improved corrosion properties, as well as their fatigue resistance [32]. Alexander Vedernikov et al. assessed the influence of pulling speed on the initial value and on the time-dependent behavior of the spring-in deformation of pultruded L-shaped structural profiles [33]. Therefore, the rapid preparation of PA56 into pultruded composites provides a research direction for our future work.

### 3.5. Flame Retardancy

The results of combustion performance demonstrated that the LOI values of PA56 and PA66 are 27.6% ± 0.9% and 22.9% ± 0.5%, respectively, and the UL94 results of both polymers are V-2. It is clear that the LOI of PA56 was much higher than that of PA66. Lyu reported that the LOI was increased to about 28% with the addition of 5 wt.% N-benzoic acid (ethyl-N-benzoic acid formamide) phosphamide (NENP) prepolymer into PA66 [34]. Li mixed about 40% *w*/*w* flame retardant composed of melamine and pentaerythritol with PA66 to increase the LOI from 22% to 27.9% [35]. In summary, the flame retardancy of PA56 was better than PA66, and PA56 can more easily undergo modifications to reach application standards regarding flame retardancy, suggesting that PA56 may have a bright future in fields that value fire resistance to a great extent.

### 3.6. Biodegradability

According to our DSC results (Table 6), the T_m_ and T_c_ of both fibers did not change much and biodegradation did not affect their melting-crystallization behavior, which was consistent with the TGA results. The T_d_^5%^ and char yield of PA56 and PA66 fibers remained almost the same for different biodegradation times (Figure 8). The main structures of both the bio-based PA56 and petroleum-based PA66 were not destroyed by the fungus eroding the fibers, which consumed the small molecules and etched the fiber surface.

The SEM images of both the PA56 and PA66 fiber surfaces during the first four weeks did not show any obvious differences since the fibers were wrapped with an oil agent and adsorbed with a small particulate (Figure 9). As the biodegradation time increased, finer fibers were gradually peeled off from the fiber surface of PA56; defects and holes appeared as well, which indicated that PA56 was degraded during the growth of phanerochaete chrysosporium. PA56 showed significant biodegradation from the eighth week, whereas PA66 remained clean and without obvious biodegradation until the sixteenth week. PA56 has denser mycelium than PA66 under the same conditions and reproduction cycle, which is more conducive to colony reproduction (Figure 10). Therefore, the biodegradation cycle of PA56 was much shorter than that of PA66. This provides an approach for the exploitation of PA56 biodegradable fiber fabrics, which is expected to contribute to the sustainability of recycling and reuse of polyamide fibers. However, this work is still inadequate because we did not explore the cause and mechanism of nylon biodegradation, which will be the focus of our future work.

## 4. Conclusions

In terms of mechanical properties, this work showed no significant differences between PA56 and PA66. The crystallization and thermal properties of PA56 in terms of melting point and decomposition temperature were lower than PA66, which suggested that PA56 could be processed with equipment similar to that used for PA66. The melt index test and rheological behavior of PA56 indicated that it is more easily processible, which will save costs and energy. PA56 might have a bright future in terms of flame retardancy because of its high LOI value (~28%). Compared to PA66, PA56 had superior biodegradability, benefiting from a shorter biodegradation cycle (8 weeks). Our systematic and comprehensive investigation indicates that PA56, based on the bio-source monomer (1,5-pentanediamine), has superior integrated performance compared to PA66, meaning that it could be a promising alternative to PA66.

## Data Availability

The data that support the findings of this study are available on request from the corresponding author.

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
