# Peer review of "The Relationships between the Structure and Properties of PA56 and PA66 and Their Fibers"

_polymers, 2023, doi:10.3390/polym15132877_

Round 1

Reviewer 2 Report

Interesting paper, well done and describied numerous experiments with PA56 and PA66 fibers. I would like to ask the Authors to make the following changes:

a) is 1,5-diamine commercialized ? There is cited Ref 11 (line 36) but this is an article, is there included address from the producer ? Is it only EPPEN Biotech (line 58) ?

b) remove space between digit number and oC, should be in line 67 310oC, the same in lines 68; 86; 88; 147; 148; 153; 154

c) please discuss a bit more difference in elongation during tensile strength measuremetns (elongation to break) - Fig. 7, lines 199 - 206. Theoretical explanation is correct but how it can influence practical applications ? Is it crucial or not at all ? Make a bit more discusssion, like this is done describing LOI in lines 218 - 219.

Round 2

Reviewer 1 Report

All major comments were adequately addressed and the Authors have done an admirable job of improving the quality of the manuscript. Therefore, it can be accepted without any structural modification.